# Long-Term Exposure of Nitrogen Oxides Air Pollution (NO_2_) Impact for Coronary Artery Lesion Progression—Pilot Study

**DOI:** 10.3390/jpm13091376

**Published:** 2023-09-14

**Authors:** Tomasz Urbanowicz, Krzysztof Skotak, Krzysztof J. Filipiak, Anna Olasińska-Wiśniewska, Krystian Szczepański, Michał Wyrwa, Jędrzej Sikora, Andrzej Tykarski, Marek Jemielity

**Affiliations:** 1Cardiac Surgery and Transplantology Department, Poznan University of Medical Sciences, 61-701 Poznan, Poland; anna.olasinska@poczta.onet.pl (A.O.-W.); mjemielity@poczta.onet.pl (M.J.); 2Institute of Environmental Protection, National Research Institute, 01-045 Warsaw, Poland; krzysztof.skotak@ios.edu.pl (K.S.); krystian.szczepanski@ios.edu.pl (K.S.); 3Institute of Clinical Science, Maria Sklodowska-Curie Medical Academy, 00-136 Warsaw, Poland; krzysztof.filipiak@uczelniamedyczna.com.pl; 4Department of Hypertensiology, Angiology and Internal Medicine, Poznań University of Medical Sciences, 61-701 Poznan, Poland; mich.wyrwa@gmail.com (M.W.); tykarski@o2.pl (A.T.); 5Poznań University of Medical Sciences, 61-701 Poznan, Poland; jedrzejsikora@gmail.com

**Keywords:** NO_2_, air pollution, CAD, PM2.5, PM10

## Abstract

Background: The potentially harmful effects of air pollution on the human health have been already presented in epidemiological studies, suggesting a strong association with increased morbidity and mortality. The aim of the study was to evaluate a possible relationship between coronary artery lesion progression related to habitation place (cities vs. villages) and air pollution. Methods: There were 148 (101 men and 47 women) patients with a median age of 70 (63–74) years enrolled into retrospective analysis based on the coronary angiography results and their habitation place. Patients with stable coronary syndrome, who underwent repeated percutaneous coronary interventions were enrolled into the analysis based on demographical and clinical characteristics combined with annual exposure to air pollution (PM2.5, PM10, and NO_2_). Results: The results of multivariable regression analysis showed a significant relationship between coronary artery lesion progression requiring percutaneous intervention and NO_2_ chronic exposure in patients living in cities of Poland (OR 2.00, 95% CI: 0.41–9.62, *p* < 0.001). The predictive value of air pollution exposure at habitation place for coronary artery lesion progression requiring percutaneous intervention was evaluated by receiver-operator curve analysis, which revealed an area under the curve of 0.939, yielding a sensitivity of 87.1% and specificity of 90.7%. Conclusions: Coronary artery lesion progression can be related to chronic exposure to NO_2_ air pollution in patients living in cities in Poland.

## 1. Introduction

Coronary artery disease is still a major cardiovascular diseases and a leading cause of mortality affecting modern population, though over the last decade, morbidity has declined [1]. A growing body of studies has investigated possible modifiable risk factors like lifestyle or physical activity that may have potential influence on patients’ prognosis [2]. The heterogeneity of coronary artery disease risk factors secondary to age-related co-morbidities were postulated by Simonetto et al. [3].

The traditional risk factors include arterial hypertension, diabetes mellitus, hyperlipidemia, homocystinuria, and psychosocial stress [4]. Overweight/obesity is considered as non-traditional and an independent risk factor for coronary artery disease development [5]. The gender differences in ischemic disease epidemiology were postulated [6,7], and gender-related differences in therapeutic approaches in the BASKET-SMALL 2 trial were presented [8]. Although more than 70% of patients referred for surgical revascularization due to multivessel disease are males [9], female gender is claimed to be associated with higher risk for greater complications and early mortality following surgery [10]. 

The potentially harmful effects of air pollution on the human health have already been presented in epidemiological studies suggesting the strong association with increased morbidity and mortality [11,12,13]. There is an escalating body of evidence indicating the role of air pollution exposure in the development of cardiovascular disease [14]. In the U.K. BioBank study, a healthy diet and annual average air pollutant concentrations of particulate matter (PM) with diameters ≤ 2.5 (PM2.5) and ≤10 (PM10) and also nitrogen oxides (NO_2_) were found related to the risk of all-cause and cardiovascular morbidity and mortality [15]. The results of the ELAPSE project revealed an association between PM2.5 exposure and increased incidence of stroke, followed by the association between coronary heart disease and nitrogen oxides [16].

Though the negative effect of air pollution on cardiovascular health is already known, the data on its relationship with the progression of coronary artery disease in individuals are lacking. The aim of the study was to evaluate the possible relationship between coronary artery lesion progression, evaluated by the need for repeated coronary angiography and angioplasty due to the occurrence of clinical symptoms, and air pollution in habitation place based on its measurements. 

## 2. Materials and Methods

There were 148 (101 men and 47 women) patients with a median age of 70 (63–74) years enrolled into the study.

This was a single-center, retrospective analysis performed based on data obtained from patients who underwent repetitive hospitalizations due to stable coronary disease syndromes between 2018 and 2022.

Demographic and clinical characteristics, including echocardiographic and laboratory results, were collected. Results of the first and consecutive coronary angiography examinations were analyzed. The atherosclerotic culprit lesion was estimated as significant for atherosclerosis on angiography as narrowing above 30%. The culprit lesion suitable for PCI intervention was defined as a stenosis above 70% or 50% in cases of non- and left-main coronary artery disease, respectively.

The progression was estimated by progression of previous mild stenosis or a new stenosis and the requirement for percutaneous coronary intervention.

Participants with acute cardiovascular syndromes and those with congestive heart failure or hematological, oncological, thyroid, liver, and kidney diseases or corticosteroid treatment were excluded from the analysis.

The basis for assessing the level of individual exposure for air pollution for particulate matter (PM) with diameters ≤ 2.5 (PM2.5) and ≤10 (PM10) and nitrogen dioxides (NO_2_) of each of the patients comprised spatial distributions of air concentration fields for Poland provided by the Chief Inspectorate of Environmental Protection [17]. The maps were based on the results for the national air quality modelling system elaborated by the Institute of Environmental Protection—National Research Institute in Poland (IEP-NRI) in accordance with the legal obligation set out in Environmental Protection Act in Poland (Art 66, paragraph 6). 

The national air quality modelling system in IEP-NRI, which is based on two procedures, namely (1) elaboration of yearly high-resolution bottom-up emission inventory for Poland and (2) elaboration air quality maps based on GEM-AQ model, operates in the Copernicus Atmosphere Monitoring Service—Regional Production (CAMS2_40) [18]. 

High-resolution bottom-up emission inventory data for Poland are developed and maintained by IEP-NRI and stored in the *Central Emission Database* [19]. All annual emissions data were elaborated based on Standard Nomenclature for Air Pollution (SNAP) categories [20], including main air pollutants emission sectors in Poland, like residential emission [21], energy production, industry, transport, or agriculture. GEM-AQ is a semi-Lagrangian chemical weather model developed at Environment Canada in which air quality processes and tropospheric chemistry are implemented in a weather prediction model, the Global Environmental Multiscale (GEM) [22]. In the GEM-AQ, the air concentration fields are performed using a 0.025-degree resolution grid. 

Personal exposure was estimated using patients’ home addresses and air quality concentration downscaling statistical methods based on an expert-in-the-loop stepwise regression procedure elaborated upon in the Neurosmog project [23], validated experimentally for real-life data from various sources aiming at predicting air pollution [24]. 

Electrocardiography (ECG) and transthoracic echocardiography (TTE) were performed in each patient before the procedures. Blood samples were collected before the procedures. Whole-blood analysis was measured with routine hematology analyzer (Sysmex Euro GmbH, Norderstedt, Germany). GFR was calculated by simplified modification of diet in renal disease (MDRD) formula.

Patients were divided into subgroups according to the habitation place. Group 1 was composed of 90 patients with a median age of 68 (57–73) years living in villages and small towns (below 50,000 citizens) compared with 58 patients with a median age of 72 (65–77) years living in a city agglomeration. The groups were matched for demographical characteristics. 

### Statistical Analysis

Continuous variables were reported as medians and interquartile ranges (Q1–Q3) since data did not follow normal distribution. Categorical data are presented as numbers and percentages. The comparison of interval parameters between proximal and non-proximal groups was performed by Mann–Whitney test. Categorical data were compared by chi-square test of independence. A logistic regression analysis was performed to identify potential predictors of coronary artery disease culprit lesion. Both univariate and multivariable models were used. The multivariable model was assessed by best subset method. The results are presented as odds ratio (OR) and its 95% confidence intervals (95%CI). Additionally, a receiver-operator characteristic (ROC) curve was determined for the predict score of the significant model including factor, which occurred as predictive in the multivariable analysis.

## 3. Results

All patients presented with chronic anginal symptoms and were hospitalized for planned coronary angiography. The first examination was performed before 2019, and second admission was carried out after 2021. The median interval between repeated examination was 1224 (680–1513) days. 

There were 114 (77%) vs. 34 (23%) patients presenting significant vs non-significant atherosclerotic lesions on initial assessment, respectively. From the mentioned 34 patients, there were 33 normal angiograms and 1 presenting lumen coronary artery stenosis < 30%.

On repeated examination from 34 patients, normal angiograms were found in 4 (12%), and non-significant progression of atherosclerotic lesions (<30%) was noticed in 6 (18%) patients. From 34 initially normal angiograms, 24 (71%) patients required percutaneous intervention on repeated examination.

On initial angiography, stent implantation was required in 109 patients; among them, the percutaneous angioplasty in 65 (60%) patients was performed during repeated angiography in comparison to 44 (40%) subjects with normal consecutive angiograms.

The majority of patients presented with co-morbidities (Table 1). Laboratory and imaging examinations results were evaluated. Both subgroups represented similar populations except for differences in left ventricular diameter and red blood cells width (RDW), which were both higher in group 2 (Table 1).

The angiographic characteristics are presented in Table 2, indicating significant difference (*p* = 0.049) in culprit lesions in repeated coronary angiography between both groups. The numbers of two-stents procedures (*p* = 0.021) and bifurcation culprit lesion (*p* = 0.023) were statistically different.

The significant differences in mean annular exposure to air pollutants between groups are presented in Table 3. 

### 3.1. Uni- and Multivariable Analysis for CAD Prediction in Patients’ Living in Cities

The univariable and multivariable analysis was performed to point out possible risk factors for culprit lesion progression and presented in Table 4. 

### 3.2. Receiver-Operator Curve Prediction for Coronary Artery Disease in Patients Living in Cities 

The predictive value of the significant factor in multivariable analysis, mean NO_2_, for coronary artery lesion progression related to air pollution exposure related to habitation place was evaluated by receiver-operator curve, revealing an area under the curve of 0.939 and yielding a sensitivity of 87.1% and specificity of 90.7%, as presented in Figure 1.

## 4. Discussion

Our study presents the relationship between coronary artery lesion disease progression and mean values of nitrogen-oxides-related air pollution in patients living in cities in one of Europe’s country regions. The analysis was performed in a country where the average density population is estimated at 117 vs. 1222/km^2^ in non-cities vs. cities areas, respectively. The results of our study revealed significant differences in polluted air components, such as PM2.5, PM10, and NO_2_, in patients living in cities in comparison to rural areas. Moreover, we found significant differences in coronary angiography results related to culprit lesion progression in patients exposed to polluted air. Chen et al. [25] in their analysis found increased mortality related to PM2.5 air pollution. In the analysis of Pasada-Sanchez et al. [26], ozone and PM2.5 exposure were found associated with premature coronary artery disease in metropolitan areas.

Not only long-term but also short-term exposure to PM2.5 and PM10 were found significant for morbidity, including lung diseases [27]. The necessity for better and more adequate air control is under continuous investigation, and new devices are under development [28].

We present the relationship between chronic coronary artery disease progression and annual exposure to air pollution. Previous studies indicated the relationship between acute coronary syndromes (ACS) and short-term exposure to air pollutants. In the study of Jiang et al. [29], a greater admission risk from acute cardiovascular events was observed under high-ozone-pollution days. Dynamic processes that occur in acute coronary syndromes that involve plaque vulnerability, fibrinolytic function, and platelet activation responsible for acute event were found related to transient exposure to environmental air in the work of Chen et al. [30]. The relationship between weather changes and increased risk for ST-segment elevated acute coronary syndromes was presented by Biondi-Zoccai et al. [31]. The impact of short-term air pollution in industrialized and non-industrialized areas on ACS was found in Kuźma et al.’s analysis [32].

Our analysis points out the significance of nitrogen oxides for coronary artery lesion progression. Kim et al. [33] in the Meta-Air study revealed an association between NO_2_ air pollution and serum lipid measurements that can explain a possible link with atherosclerosis progression. Tian et al. [34] in their study presented an association with the transition from healthy status to incident stroke with each 5 μg/m^3^ increase in nitrogen oxides in the air. Dominski et al. [35] in their review pointed out the primary relationship between air pollution and respiratory diseases (mainly asthma and COPD), followed by cardiovascular outcomes (mainly stroke).

It is widely known that over 90% of the pollutant mass is represented by the mixture of ozone, nitrogen oxide, volatile organic compounds, and sulfur dioxide (SO_2_) [36]. Previous reviews presented the pathophysiological mechanisms that link atherosclerosis progression and explained increased acute coronary syndromes risk in patients exposed to PM2.5 [37]. Air’s polluted particles are claimed to interfere with endothelial function, activate cytokines, and derange lipid profiles to provoke atherosclerosis progression [38,39]. Moreover, the Zhang et al. [40] presented the increased risk for secondary acute coronary syndrome among patients with short- or long-term air pollution exposure, including nitrogen oxide.

Our study points out the significance of nitrogen oxide in comparison to previous reports that the main constituent of highly polluted air is particulate matter, which is claimed as responsible for the production of reactive oxygen radicals and the alteration of calcium levels [41]. Chen et al. [42] in their recent analysis found the relationship between NO exposure and subclinical atherosclerosis among young adults. Most importantly, Brunekreef et al. [43] in their meta-analysis found an association between PM2.5 and NO long-term exposure and increased all-cause mortality that was below current limits. 

The presented results from our analysis correlate nitric oxide air pollution with chronic coronary syndrome, indicating progressive atherosclerotic plaque development. This novel finding broadens the scientific perspective on the pathophysiology of coronary culprit lesion and, in contrary to previous reports, suggests the relationship between meteorological factors and acute coronary syndrome risk [44,45]. More interestingly, Diaz-Chiron et al. [46] presented the results of myocardial infarct size in relation to nitric oxide air and pollution and inflammatory activation measured by the neutrophil-to-lymphocyte ratio. It is worth adding that in our analysis, none of the inflammatory indexes obtained from whole-blood-count analysis showed such a correlation. 

Nitrogen oxides pollution was found to be influenced by road transport and the industrial combustion and processes sectors [47]. Kiesseweter et al. in their analysis attributed the contribution of increased NO_2_ concentrations to the high shares of diesel cars [48]. Newell et al. [49] in their meta-analysis found the association between nitric oxide and chronic obstructive pulmonary disease (COPD), followed by increased mortality. 

In our analysis, the mean annual NO_2_ exposure was above the levels suggested by World Health Organization [50]. Its seasonal variations were presented by Wang et al. [51] as a possible morbidity risk factor related to maximal exposure, as Chen et al. found a relationship between hourly air pollution changes and risk for acute coronary syndromes [30]. The significant relationship between all-cause mortality and air pollution, especially when combined with increased concentrations of PM2.5 and nitric oxide, was presented in Liu et al.’s study [52].

### Study Limitation

The study was performed on a relatively low-volume population and assumed that patients did not change their habitation place for a long period of time (except for short-term vacations), according to interview protocol. The study was based on one region in an EU country that accounts for 3.5 million citizens.

## 5. Conclusions

Coronary artery lesion progression can be related to chronic exposure to NO_2_ air pollution in patients living in cities in Poland.

## Figures and Tables

**Figure 1 jpm-13-01376-f001:**
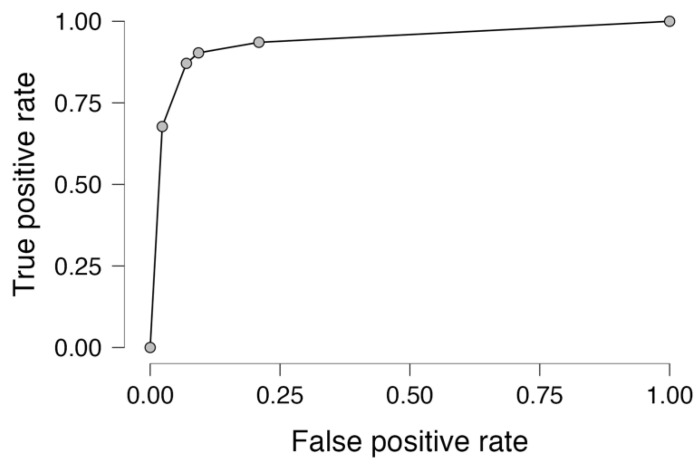
Receiver-operator curve for mean NO_2_ exposure in coronary artery disease patients’ living in cities.

**Table 1 jpm-13-01376-t001:** Clinical and demographical characteristics.

Parameter	Group 1	Group 2	*p*
	n = 90	n = 58	
Sex (male (%)/female (%))	66 (73)/24 (27)	36 (62)/22 (38)	0.657
Age (years) (median (Q1–Q3))	68 (57–73)	72 (65–77)	0.074
Weight (kg) (median (Q1–Q3))	93 (92–94)	90 (67–94)	0.434
Height (cm) (median (Q1–Q3))	168 (162–175)	171 (161–176)	0.703
Body mass index (median (Q1–Q3))	29.7 (27.2–33.8)	29.1 (25.7–31.5)	0.398
Co-morbidities:			
Arterial hypertension (n(%))	47 (52)	34 (59)	0.445
Diabetes mellitus (n(%))	26 (29)	14 (24)	0.525
Hypercholesterolemia (n(%))	48 (53)	30 (52)	0.848
COPD (n(%))	5 (6)	3 (5)	0.919
Thyroid disease (n(%))	10 (11)	5 (9)	0.624
Atrial fibrillation (n(%))	8 (9)	4 (7)	0.665
Current nicotinism (n(%))	3 (3)	1 (2)	0.556
Stroke (n(%))	3 (3)	3 (5)	0.58
PAD (n(%))	9 (10)	5 (9)	0.78
Current nicotinism (n(%))	3 (3)	1 (2)	0.556
Echocardiographic results			
Left ventricular diameter (mm) (median (Q1–Q3))	40 (38–43)	47 (44–50)	0.044 *
Left atrium diameter (mm) (median (Q1–Q3))	35 (30–37)	39 (37–42)	0.137
Left ventricular ejection fraction (%)(median (Q1–Q3))	65 (60–65)	55 (55–60)	0.199
Laboratory results			
WBC (×10^9^/L) (median (Q1–Q3))	7.48 (7.29–8.41)	7.44 (6.13–8.67)	0.669
Neutrophils (×10^9^/L) (median (Q1–Q3))	5.28 (4.90–5.64)	4.14 (3.54–5.58)	0.269
Lymphocyte (×10^9^/L) (median (Q1–Q3))	1.70 (1.54–2.11)	1.98 (1.45–2.20)	0.88
Monocyte (×10^9^/L) (median (Q1–Q3))	0.50 (0.42–0.52)	0.46 (0.41–0.57)	0.801
Platelets (×10^9^/L) (median (Q1–Q3))	218 (216–231)	218 (178–255)	0.88
Hemoglobin (mmol/L) (median (Q1–Q3))	7.70 (7.55–8.30)	8.60 (8.25–8.98)	0.19
Hematocrit (%) (median (Q1–Q3))	36 (35–40)	38 (34–43)	0.21
MPV (fl) (median (Q1–Q3))	9.1 (8.5–9.4)	8.4 (7.9–9.2)	0.58
MCV (median (Q1–Q3))	90 (89–92)	93 (90–97)	0.313
MCHC (mmol/L) (median (Q1–Q3))	20.87 (20.82–20.97)	20.96 (20.59–21.40)	0.802
RDW (%) (median (Q1–Q3))	12.60 (12.55–12.90)	13.80 (13.18–14.18)	0.030 *
Creatinine (median (Q1–Q3))	90 (83–108)	92 (71–108)	0.725
GFR (umol/l) (median (Q1–Q3))	59 (57–65)	74 (60–86)	0.242
Ureic acid (median (Q1–Q3))	296 (272–352)	363 (317–403)	0.634

Abbreviations: COPD, chronic obstructive pulmonary disease; GFR, glomerular filtration rate; MCHC, mean corpuscular hemoglobin concentration; MCV, mean corpuscular volume; MPV, mean platelets volume; PAD, peripheral artery disease; RDW, red cells distribution width; * statistically significant.

**Table 2 jpm-13-01376-t002:** Comparison of coronary angiographic results within both groups.

	Group 1	Group 2	*p*
	n = 90	n = 58	
First angiographic results			
1. Disease (>30% stenosis):			
1. LMCA disease (n,%)	6 (7)	4 (7)	0.96
2. LAD disease (n,%)	52 (58)	33 (57)	0.918
3. Cx disease (n,%)	29 (32)	16 (28)	0.552
4. RCA disease (n,%)	42 (47)	23 (40)	0.404
2. Normal angiogram (n,%)	21 (23)	11 (19)	0.532
3. Culprit lesion > 70% (n,%)	75 (83)	56 (97)	0.179
4. Performed procedures:			
1. Single PCIs (n,%)	55 (61)	37 (64)	0.645
2. Two stents (n,%)	10 (11)	7 (12)	0.773
3. On bifurcation (n,%)	5 (6)	5 (9)	0.84
Second angiographic results			
1. Disease (>30% stenosis):			
1. LMCA disease (n,%)	7 (8)	5 (9)	0.858
2. LAD disease (n,%)	41 (46)	26 (45)	0.933
3. Cx disease (n,%)	29 (32)	11 (19)	0.078
4. RCA disease (n,%)	33 (37)	26 (45)	0.325
2. Normal angiogram (n,%)	3 (3)	1 (2)	0.079
3. Culprit lesion >70% (n,%)	46 (51)	55 (9)	0.049 *
4. Performed procedures:			
1. Single PCIs (n,%)	44 (49)	26 (45)	0.732
2. Two stents (n,%)	1 (1)	11 (19)	0.021 *
3. Bifurcation (n,%)	0 (0)	7 (12)	0.023 *
5. Culprit lesion change (%)	55 (10–90)	68 (15–97)	0.383

Abbreviations: Cx, circumflex artery; LAD, left descending artery; LMCA, left main coronary artery; PCI, percutaneous intervention; RCA, right coronary artery. * statistically significant.

**Table 3 jpm-13-01376-t003:** Exposure to air pollution.

Parameter	Group 1	Group 2	*p*
	n = 90	n = 58	
Particulate Matter < 2.5 (PM < 2.5)			
Mean exposure in 2019 (median (Q1–Q3))	17.5 (15.5–19.1)	17.5 (16.6–18.7)	0.736
Mean exposure in 2020 (median (Q1–Q3))	12.6 (11.0–15.1)	16.4 (14.7–19.1)	<0.001
Mean exposure in 2021 (median (Q1–Q3))	14.6 (13.4–16.3)	17.7 (15.8–19.2)	<0.001
Mean exposure 2019–2021 (median (Q1–Q3))	15.8 (13.6–16.6)	17.2 (16.0–18.9)	<0.001
Maximal exposure (median (Q1–Q3))	18 (15.5–19.3)	18.9 (16.9–20.0)	0.269
Particulate Matter < 10 (PM < 10)			
Mean exposure in 2019 (median (Q1–Q3))	23.4 (21.8–25.0)	26.4 (24.6 28.6)	<0.001
Mean exposure in 2020 (median (Q1–Q3))	20.2 (18.4–22.0)	23.7 (21.9–28.1)	<0.001
Mean exposure in 2021 (median (Q1–Q3))	23.3 (22.1–28.1)	25.9 (23.6–29.7)	<0.001
Mean exposure 2019–2021 (median (Q1–Q3))	22.2 (21.5–23.5)	25.4 (23.6 28.7)	<0.001
Maximal exposure (median (Q1–Q3))	24.0 (22.6–26.00)	26.6 (25.2–30.1)	<0.001
Nitrogen Oxides (NO_2_)			
Mean exposure in 2019 (median (Q1–Q3))	12.2 (10.5–14.2)	22.4 (19.4–24.00)	<0.001
Mean exposure in 2020 (median (Q1–Q3))	11.2 (10.2–13.1)	17.8 (16.5–18.7)	<0.001
Mean exposure in 2021 (median (Q1–Q3))	12.6 (12.00–13.8)	18.1 (17.0–19.5)	<0.001
Mean exposure 2019–2021 (median (Q1–Q3))	12.1 (11.0–13.6)	19.4 (17.8–20.7)	<0.001
Maximal exposure (median (Q1–Q3))	13.0 (12.0–14.2)	22.5 (19.7–24.0)	<0.001

**Table 4 jpm-13-01376-t004:** Univariable and multivariable analysis for NO_2_ exposure related to CAD progression.

	Univariable			Multivariable		
	OR	95% CI	*p*	OR	95% CI	*p*
Demographical factors:				-	-	-
Sex	1.24	0.42–1.84	<0.001 *			
Age	1.05	0.10–1.13	0.055			
Co-morbidities				-	-	-
Arterial hypertension	5.43	0.42–29.63	0.009 *			
DM	2.83	0.01–2.92	0.073			
Hyperlipidemia	2.65	0.50–14.55	<0.001 *			
Thyroid disease	1.99	0.14–1.15	0.418			
Laboratory results:				-	-	-
WBC	0.66	0.03–0.21	0.197			
Hemoglobin	2.06	0.61–2.05	0.286			
NLR	1.09	0.77–1.09	0.557			
MLR	4.83	1.67–4.82	0.342			
SIRI	2.23	1.14–2.74	0.42			
MCHC	0.77	0.61–2.43	0.243			
Creatinine	1.01	0.03–0.53	0.486			
Ureic acid	1	0.03–1.02	0.734			
Air pollution—PM < 2.5				-	-	-
PM < 2.5 in 2019	0.98	0.796–1.722	0.859			
PM < 2.5 in 2020	4.13	0.23–6.42	<0.001			
PM < 2.5 in 2021	1.53	0.20–6.58	<0.001			
Mean PM < 2.5 exposure	1.44	0.21–5.17	<0.001			
Max PM < 2.5 exposure	1.15	0.01–2.76	0.033			
Air pollution—PM < 10				-	-	-
PM < 10 in 2019	1.39	0.20–4.55	<0.001			
PM < 10 in 2020	1.43	0.23–4.87	<0.001			
PM < 10 in 2021	1.28	0.13–3.55	<0.001			
Mean PM < 10 exposure	1.44	0.23–5.01	<0.001			
Max PM < 10 exposure	1.33	0.17–4.04	<0.001			
Air pollution—NO_2_						
NO_2_ in 2019	1.57	0.33–5.80	<0.001			
NO_2_ in 2020	1.99	0.49–8.80	<0.001			
NO_2_ in 2021	2.35	0.61–1.10	<0.001			
Mean	1.93	0.48–8.45	<0.001	2	0.41–9.62	<0.001
Max	1.76	0.41–7.27	<0.001			

Abbreviations: DM, diabetes mellitus; max, maximum; MCHC, mean corpuscular hemoglobin concentration; NO_2_, nitrogen oxides; OR, odds ratio; RDW, red cells distribution width; WBC, white blood count. * statistically significant.

## Data Availability

All data will be available for 3 years following the publication after reasonable request is presented in e-mail correspondence to the corresponding author.

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
