# Peer review of "Long-Term Exposure of Nitrogen Oxides Air Pollution (NO2) Impact for Coronary Artery Lesion Progression—Pilot Study"

_jpm, 2023, doi:10.3390/jpm13091376_

Round 1

Reviewer 1 Report

I've read with interest the paper entitled "long term exposure of nitrogen oxides air polluttion (NO2) 2 impact for coronary artery disease progression – pilot study" but I'm a little bit confused. Material and methods are summarised in 6 lines and the design of the study is not clear at all. How was coronary artery disease "quantity" evaluated? And the title speaks about progression: how was the progression estimated? a ROC curve of 0.93 is excellent but it is not clear at all how it was calculated.

I would therefore suggest to explain better the protocol and try to improve the text in order to clarify the topic of coronary artery disease evaluation and progression.

English should be improved and some minor changes also made.

Example: 

25 annual not annular) exposure to air pollution (PM2.5, PM 10 and NO2)

48 The long-term results of interventions due to 48 coronary disease are sex-related [9.10.11] are postulated. ?????????what does it mean??

97 There were 17 (6 vs 11) angiograms on initial examination. ??????? what does it mean??

Should be improved

Author Response

Rev 1

We would like to take this opportunity to express our gratitude to the Reviewer for taking valuable time to review our manuscript as well as the highly encouraging comments. In compliance with the professional suggestions of the Reviewer, we have revised our manuscript accordingly. Please find our point-by-point responses in an attached file. Please also kindly note that the corresponding changes in the revised manuscript are marked in red.  

I've read with interest the paper entitled "long term exposure of nitrogen oxides air polluttion (NO2) 2 impact for coronary artery disease progression – pilot study" but I'm a little bit confused.

  1. Material and methods are summarised in 6 lines and the design of the study is not clear at all.

Material and methods section was corrected with the description of inclusion and exclusion criteria, methods of laboratory examinations, methods of air pollution measurement. Study design was clarified.

  1. How was coronary artery disease "quantity" evaluated?

The atherosclerotic culprit lesion was estimated as significant for atherosclerosis on angiography as narrowing above 30%. The culprit lesion suitable for PCI intervention was above 70%.

  1. And the title speaks about progression: how was the progression estimated?

The progression was estimated by progression of previous mild or none stenosis or a new stenosis and the requirement for percutaneous coronary intervention.

The predictive value for coronary artery disease progression requiring percutaneus intervention related to air-pollution exposure at habitation place was evaluated by receiver operator curve analysis that revealed area under the curve 0.939 yielding the sensitivity of 87.1% and specificity of 90.7%.

  1. a ROC curve of 0.93 is excellent but it is not clear at all how it was calculated.

The ROC curve analysis was performed based on the factor significant in the multivariable analysis (Mean NO2).

I would therefore suggest to explain better the protocol and try to improve the text in order to clarify the topic of coronary artery disease evaluation and progression.

                        We did our best to improve the text of Methods and results sections.

The basis for assessing the level of individual exposure for air pollution for paticulatte matter (PM) with diameters ≤2.5 (PM2.5) and ≤10 (PM10) and nitrogen dioxides (NO2) of each of the patients, were spatial distributions of air concentration fields for Poland provided by the Chief Inspectorate of Environmental Protection []. The maps were based on the results for the national air quality modelling system elaborated by Institute of Environmental Protection – National Research Institute in Poland (IEP-NRI) in accordance with the legal obligation set out in Environmental Protection Act in Poland (Art 66, paragraph 6).

The national air quality modelling system in IEP-NRI based on the two procedures: (1) elaboration of yearly high-resolution bottom-up emission inventory for Poland and (2) elaboration air quality maps based on GEM-AQ model, operates in the Copernicus Atmosphere Monitoring Service—Regional Production (CAMS2_40) [].

High-resolution bottom-up emission inventory data for Poland are developed and maintained by IEP-NRI and stored in the Central Emission Database []. All annual emissions data are elaborated based on Standard Nomenclature for Air Pollution (SNAP) categories [] including main air pollutants emission sectors in Poland like residential emission [], energy production, industry, transport or agriculture. GEM-AQ is a semi-Lagrangian chemical weather model developed at Environment Canada in which air quality processes and tropospheric chemistry are implemented in a weather prediction model, the Global Environmental Multiscale (GEM) []. In the GEM-AQ, the air concentration fields are performed using a 0.025-degree resolution grid.

Personal exposure was estimated using patients’ home address and air quality concentration downscaling statistical methods based on Expert-in-the-loop Stepwise Regression procedure elaborated in Neurosmog project [], validated experimentally for real-life data from various sources aiming at predicting air pollution [].

English should be improved and some minor changes also made.

We corrected the text

Example: 

25 annual not annular) exposure to air pollution (PM2.5, PM 10 and NO2)

Corrected

48 The long-term results of interventions due to 48 coronary disease are sex-related [9.10.11] are postulated. ?????????what does it mean??

The gender differences in ischemic disease epidemiology are postulated [] and sex related differences in therapeutic approaches in BASKET-SMALL 2 trial were presented related to sex disparities  []

97 There were 17 (6 vs 11) angiograms on initial examination. ??????? what does it mean??

We clarified the context:

There were 114 (77%) vs 34 (23%) patients presenting significant vs non-significant atherosclerotic lesions on initial assessment, respectively. From mentioned 34 patients, there were 33 normal angiograms and 1 presenting lumen coronary artery stenosis < 30%.

On repeated examination from 34 patients, the normal angiograms were found in 4 (12%), non-significant progression of atherosclerotic lesions (<30%) was noticed in 6 (18%) patients. From 34 initially normal angiograms, 24 (71%) patients required percutaneous intervention on repeated examination.

On initial angiography, stent implantation was required in 109 patients; among them, the percutaneous angioplasty in 65 (60%) patients was performed during repeated angiography in contrary to 44 (40%) subjects with normal consecutive angiograms.

Majority of patients presented with co-morbidities (Table 1). Laboratory and imaging examinations results were evaluated. Both subgroups represented similar populations, beside differences in left ventricular diameter and red blood cells width (RDW) which were both higher in group 2 (Table 1).

Kind regards

Tomasz Urbanowicz on behalf of all co-authors

Reviewer 2 Report

The research represents an epidemiological study, aiming to evaluate association between the exposure to air pollutants and progression of coronary artery disease.

There are certain concerns that need to be addressed:

1. Materials and methods need to be corrected. Information on how information about patients was gathered is missing. Dates when the study was performed? Exclusion criteria? Which centers were involved? Were any data bases used for the analyses? Methods do not contain information on how air pollution exposure was analyzed? Study design as well may be represented here.

Some recent examples of methods for epidemiological studies include but are not limited to:

https://www.mdpi.com/2079-7737/11/8/1122

https://pubmed.ncbi.nlm.nih.gov/36554535/

2. Table 1 should be in results. It is more appropriate to represent not weight and height separately, but body mass index (BMI) calculated basing on these parameters.

What does asterisk indicate in Table 1? If p level<0.05, why % of arterial hypertension is not marked. Anyway, it should be deciphered below the table.

3. Why in table 1 hematocrit is not represented for group 2?

4. In results: “The atherosclerotic culprit lesion was estimated as significant for 98 atherosclerosis on angiography as narrowing above 30%” – why in the table 2 the presence of disease is indicated as >40% stenosis?

5. How the fact of progression of coronary artery disease was stated? Overall, how many patients in each group had the progression of atherosclerosis? As far as I understood, this was dependent variable in the models, not the performed procedures, or presence of culprit lesion>70%, etc. Does ROC curve characterize the multivariable model? It is not clear from the text.  

7. The role of one of coauthors was funding acquisition. Why is it stated that research received no external funding?

8. The date of approval of the study protocol is not indicated (Line 221).

9. There are grammar mistakes through the text: time tenses require authors’ attention. There a mistake in the heading: should be air pollution.  

There are grammar mistakes through the text. English needs to be edited. 

Author Response

We would like to take this opportunity to express our gratitude to the Reviewer for taking valuable time to review our manuscript as well as the highly encouraging comments. In compliance with the professional suggestions of the Reviewer, we have revised our manuscript accordingly. Please find our point-by-point responses in attached file. Please also kindly note that the corresponding changes in the revised manuscript are marked in red.  

The research represents an epidemiological study, aiming to evaluate association between the exposure to air pollutants and progression of coronary artery disease. 

There are certain concerns that need to be addressed:

  1. Materials and methods need to be corrected.

Material and methods section was corrected with the description of inclusion and exclusion criteria, methods of laboratory examinations, methods of air pollution measurement. Study design was clarified.

  1. Information on how information about patients was gathered is missing.

Patients with stable coronary syndrome, who underwent repeated hospitalizations were enrolled into the analysis.

  1. Dates when the study was performed?

The information was added.

The single center retrospective analysis was performed basing on information obtained from patients who underwent repetitive hospitalization due to stable coronary disease syndromes between 2018 and 2022.

  1. Exclusion criteria?

Participants with acute cardiovascular syndromes and those with congestive heart failure, hematological, oncological, thyroid, liver and kidney disease or corticosteroid treatment were excluded from the analysis.

  1. Which centers were involved?

There was a one center analysis. The center is covering the region of 3,5 million citizens.

  1. Were any data bases used for the analyses?

Information presenting air pollution exposure were obtained from database of Institute of Envirolmental Protection – National Research Institute in Poland and was calculated individually for every patient in relation to his habitation.

  1. Methods do not contain information on how air pollution exposure was analyzed?

Information presenting air pollution exposure were obtained from database of Institute of Envirolmental Protection – National Research Institute in Poland and was calculated individually for every patient in relation to his habitation.

  1. Study design as well may be represented here.

Some recent examples of methods for epidemiological studies include but are not limited to: 

https://www.mdpi.com/2079-7737/11/8/1122

https://pubmed.ncbi.nlm.nih.gov/36554535/

The methods were corrected and the results of both studies were mentioned in discussion.

  1. Table 1 should be in results. It is more appropriate to represent not weight and height separately, but body mass index (BMI) calculated basing on these parameters.

We corrected Table 1

What does asterisk indicate in Table 1? If p level<0.05, why % of arterial hypertension is not marked. Anyway, it should be deciphered below the table.

* means significant statistically, we added the information. % of hypertension was checked and corrected.

  1. Why in table 1 hematocrit is not represented for group 2?

                        It was corrected.

  1. In results: “The atherosclerotic culprit lesion was estimated as significant for 98 atherosclerosis on angiography as narrowing above 30%” – why in the table 2 the presence of disease is indicated as >40% stenosis? 

                        It was digit mistake and was corrected.

  1. How the fact of progression of coronary artery disease was stated? Overall, how many patients in each group had the progression of atherosclerosis? As far as I understood, this was dependent variable in the models, not the performed procedures, or presence of culprit lesion>70%, etc. Does ROC curve characterize the multivariable model? It is not clear from the text.  

The progression was estimated by progression of previous mild stenosis, or a new stenosis and the requirement for percutaneous coronary intervention due to artery lumen narrowing at least 70% or more than 50% stenosis in left main coronary artery cases.

Yes, the progression was a dependent variable.

  1. The role of one of coauthors was funding acquisition. Why is it stated that research received no external funding?

The rest of this author roles was missed and it was corrected, we ALSO  considered the person responsible for future APC payment.

  1. The date of approval of the study protocol is not indicated (Line 221).

Corrected

The study was conducted in accordance with the Declaration of Helsinki and approved by the Institutional Review Board (or Ethics Committee) of Poznan University of Medical Sciences, Poznan, Poland (protocol code 55/20 from 16 January 2020).” for studies involving humans.

  1. There are grammar mistakes through the text: time tenses require authors’ attention. There is a mistake in the heading: should be air pollution.  

                        Corrected.

Kind regards

Tomasz Urbanowicz

nn behalf of all-authors

Round 2

Reviewer 1 Report

I've read with attention and interest the submitted paper. Air pollution is described exhaustively. Unfortunately the authors are not able to describe coronary artery disease in an acceptable way. The method used is not recognised. They state that all the patients had chronic artery disease and  that patients with acute coronary artery disease were excluded. I believe that the best demonstration of atherotic burden progression is the development of a unstable situation like an acute coronary event. So I don't understand why ASC patients were excluded. Furthermore, if the authors wanted to concentrate only on chronic coronary lesions, they should have used a recognised method to classify degree and diffusion of the atheroterosclerotic problem as for example the SYNTAX Score. Even an apparently small diffused increase of atheroscleorosis is a progression and not only the need of hospital readmission or PCI performance. Why were patients with chronic artery disease readmitted? Why did they undergo PCI? The indications to readmission or PCI are subjective and could not be considered a proof of progression.

The need of coronary angiogram or PCI due to occorence of clinical symptoms: new clinical symptoms? In that case it is not a chronic problem. Stable clinical symptoms? So what was the reason of proceding with a new angiogram or PCI?

I think the idea of correlating air pollution with coronary atherosclerotic disease progression is excellent but it should be demonstrated using adequate methods. The AUC of 0.93 is striking: but it is based on a wrong assumption: ACS excluded from progression and artery disease degree extimated in a wrong way.

Some other minor shortcomings of the article:

english is often embarassing: Line 17 asbtract: has been not has/Text line 48: what's the sense of the sentence ... The long term ..???/.....

Line 67 the need for repeated coronary angiography and angioplasty due to the occurrence of clinical symptoms. It is not a way to evaluate coronary artery disease progression

Line 75: the authors say "the patients underwent repeated hospitalisation due to stable coronary artery disease. So if the cause of hsospitalisation was stable coronary artery disease it could not be clinically afforded as progression

LIne 83: The progression was estimated by progression of previous mild stenosis, or a new stenosis and the requirement for percutaneous coronary intervention. It is not a way to evaluate angiographic progression. A progression of a mild stenosis could simply depend of a different operator who classified the stenosis in different way. It is well known that a single lesion could be classified differently by different operators or even by the same operator in different time

86.87 Participants with acute cardiovascular syndromes and those with congestive heart failure, hematological, oncological, thyroid, liver and kidney diseases or corticosteroid treatment were excluded from the analysis. It is strange to esclude patients with acute coronary disease. Isn't it a progression from a chronic condition?

118 electrocardiography (ECG) and transthoracic echocardiography (TTE) were performed in each patient before the procedures. Blood samples were collected before the procedures. Whole blood analysis was measured with routine hematology analyser (Sysmex Euro GmbH, Norderstedt, Germany). GFR was calculated by simplified Modification Diet in Renal Disease (MDRD) formula. Completely useless informations

Other risk factor? WHat about cholesterol? Were the findings normalised for cholesterol levels?

How was the rate of progression? If a lesion increase from 30 to 60% over a period of 5 years or from 80 to 90% over a period of 3 years what's worse?

Should be improved

Author Response

Poznan, 9/9/2023

Dear Reviewer,

We would like to take this opportunity to express our gratitude to the Reviewer for taking valuable time to review our manuscript as well as the encouraging comments. Please find our point-by-point responses in attached file.

I've read with attention and interest the submitted paper. Air pollution is described exhaustively. Unfortunately the authors are not able to describe coronary artery disease in an acceptable way. The method used is not recognised. They state that all the patients had chronic artery disease and  that patients with acute coronary artery disease were excluded. I believe that the best demonstration of atherotic burden progression is the development of a unstable situation like an acute coronary event. So I don't understand why ASC patients were excluded.

Dear Reviewer, we would like to thank you for your comment. The analysis was performed basing on annual mean values of air-pollution not the maximal daily results. The mean annual values expressed the chronic exposure that can be related to chronic progression not to acute coronary syndromes. If the short-term variations of air-pollutant would be measured, the analysis of the acute syndromes would be justified.

Furthermore, if the authors wanted to concentrate only on chronic coronary lesions, they should have used a recognised method to classify degree and diffusion of the atheroterosclerotic problem as for example the SYNTAX Score. Even an apparently small diffused increase of atheroscleorosis is a progression and not only the need of hospital readmission or PCI performance.

Dear Reviewer, we appreciate your comment but please let us present the answer. The analysis was based on angiographic results including the coronary arteries lumen stenosis. The atherosclerotic lesions were measured in epicardial arteries according to coronary angiograms. Not the SYNTAX score that present the complexity of the culprit lesion but the percentage of arteries that was followed on repeated angiography was the aim of the study. Including the SYNTAX score into analysis could be misleading. The analysis did not focused on small arteries disease as the angiograms present the epicardial artery disease.

Why were patients with chronic artery disease readmitted? Why did they undergo PCI?

Dear Reviewer, we would like to thank you for your comment. All patients were readmitted due to clinically expressed chronic coronary syndromes. The PCI was performed when the stenosis of the coronary lumen was revealed on angiograms >50% and >70% in left main and other coronary arteries, respectively.

The indications to readmission or PCI are subjective and could not be considered a proof of progression.

Dear Reviewer, we would like to thank you for your comment. The readmission was based on cardiologist decision that referred patients for repeated hospitalization and confirmed by experienced clinician in Clinical Department that we can’t argue with. The experienced two physician confirmed the justification for the procures before the angiography was performed.

The need of coronary angiogram or PCI due to occorence of clinical symptoms: new clinical symptoms? In that case it is not a chronic problem. Stable clinical symptoms? So what was the reason of proceding with a new angiogram or PCI?

As we said the, patients presenting with chronic syndromes that limit their life activity and are not combined by ECG ischemic changes nor myocardial ischemia markers increase were the primary indication for rehospitalization and repeated angiography.

I think the idea of correlating air pollution with coronary atherosclerotic disease progression is excellent but it should be demonstrated using adequate methods. The AUC of 0.93 is striking: but it is based on a wrong assumption: ACS excluded from progression and artery disease degree extimated in a wrong way.

Dear Reviewer, we would like to thank you for your opinion, but we can’t agree with your suggestion as the mean annual air-polution exposure was measured. The mean annual values can be related to chronic changes not to acute syndromes. Any acute clinical scenario is following acute trigger not the chronic one, especially that is measured annually.

As you may find in enclosed publications, the air-pollution acute changes were related to acute syndromes:

Zhang H, Yin L, Zhang Y, Qiu Z, Peng S, Wang Z, Sun B, Ding J, Liu J, Du K, Wang M, Sun Y, Chen J, Zhao H, Song T, Sun Y. Short-term effects of air pollution and weather changes on the occurrence of acute aortic dissection in a cold region. Front Public Health. 2023 Aug 2;11:1172532. doi: 10.3389/fpubh.2023.1172532. PMID: 37601173; PMCID: PMC10433911.

Jiang Y, Huang J, Li G, Wang W, Wang K, Wang J, Wei C, Li Y, Deng F, Baccarelli AA, Guo X, Wu S. Ozone pollution and hospital admissions for cardiovascular events. Eur Heart J. 2023 May 7;44(18):1622-1632. doi: 10.1093/eurheartj/ehad091. PMID: 36893798.

Mills NL, Pope CA. Environmental Triggers of Acute Coronary Syndromes. Circulation. 2022 Jun 14;145(24):1761-1763. doi: 10.1161/CIRCULATIONAHA.122.059861. Epub 2022 Jun 13. PMID: 35696455.

Pope CA, Muhlestein JB, Anderson JL, Cannon JB, Hales NM, Meredith KG, Le V, Horne BD. Short-Term Exposure to Fine Particulate Matter Air Pollution Is Preferentially Associated With the Risk of ST-Segment Elevation Acute Coronary Events. J Am Heart Assoc. 2015 Dec 8;4(12):e002506. doi: 10.1161/JAHA.115.002506. PMID: 26645834; PMCID: PMC4845284.

Seasonal but not annual relation was presented between air-pollution and ACS in enclosed publication:

Ong GJ, Sellers A, Mahadavan G, Nguyen TH, Worthley MI, Chew DP, Horowitz JD. 'Bushfire Season' in Australia: Determinants of Increases in Risk of Acute Coronary Syndromes and Takotsubo Syndrome. Am J Med. 2023 Jan;136(1):88-95. doi: 10.1016/j.amjmed.2022.08.013. Epub 2022 Sep 2. PMID: 36058309.

The daily mean values of the air pollutants from the day before until 7 days before admission were analysed in Laura Diaz-Chiron study: Díaz-Chirón L, Negral L, Megido L, Suárez-Peña B, Domínguez-Rodríguez A, Rodríguez S, Abreu-Gonzalez P, Pascual I, Moris C, Avanzas P. Relationship Between Exposure to Sulphur Dioxide Air Pollution, White Cell Inflammatory Biomarkers and Enzymatic Infarct Size in Patients With ST-segment Elevation Acute Coronary Syndromes. Eur Cardiol. 2021 Dec 7;16:e50. doi: 10.15420/ecr.2021.37. PMID: 34950246; PMCID: PMC8674636.

Biondi-Zoccai G, Frati G, Gaspardone A, Mariano E, Di Giosa AD, Bolignano A, Dei Giudici A, Calcagno S, Scappaticci M, Sciarretta S, Valenti V, Casati R, Visconti G, Penco M, Giannico MB, Peruzzi M, Cavarretta E, Budassi S, Cosma J, Federici M, Roever L, Romeo F, Versaci F. Impact of environmental pollution and weather changes on the incidence of ST-elevation myocardial infarction. Eur J Prev Cardiol. 2021 Oct 25;28(13):1501-1507. doi: 10.1177/2047487320928450. Epub 2020 Jun 2. PMID: 34695216.

Kuźma Ł, Wańha W, Kralisz P, Kazmierski M, Bachórzewska-Gajewska H, Wojakowski W, Dobrzycki S. Impact of short-term air pollution exposure on acute coronary syndrome in two cohorts of industrial and non-industrial areas: A time series regression with 6,000,000 person-years of follow-up (ACS - Air Pollution Study). Environ Res. 2021 Jun;197:111154. doi: 10.1016/j.envres.2021.111154. Epub 2021 Apr 17. PMID: 33872649.

Finally we added the information into the manuscript:

We present the relation between chronic coronary artery disease progression and annual exposure to air-pollution. Previous studies indicated the relation between acute coronary syndromes (ACS) and short-term exposure to air pollutants. In Jiang et al. [[i]] analysis the greater admission risk from acute cardiovascular events was observed under high ozone pollution days. Dynamic process that occurs in acute coronary syndromes that involve plaque vulnerability, fibrinolytic function, and platelet activation responsible for acute event were found related to transient exposure to environmental air by Chen et al []. The relation between weather changes and increased risk for ST-segment elevated acute coronary syndromes were presented by Biondi-Zoccai et al. []. The impact of short-term air pollution in industrialized and non-industries areas on ACS was already found in Kuźma et al. analysis [].

Some other minor shortcomings of the article:

english is often embarassing: Line 17 asbtract: has been not has/Text line 48: what's the sense of the sentence ... The long term ..???/.....

Thank you for valuable comments. It was corrected and the sentence was erased.

Line 67 the need for repeated coronary angiography and angioplasty due to the occurrence of clinical symptoms. It is not a way to evaluate coronary artery disease progression

Dear Reviewer, thank you for you comment. Patients with stable but progressive symptoms after thorough cardiologist examinations are referred for repeated angiographies as was presented in our analysis. As we stated, the two professional clinicians were involved in final decision regarding coronary angiography.

Line 75: the authors say "the patients underwent repeated hospitalisation due to stable coronary artery disease. So if the cause of hsospitalisation was stable coronary artery disease it could not be clinically afforded as progression

Dear Reviewer thank you for your valuable comment but I you have read our manuscript; the progression was estimated by coronary lumen narrowing and was presented in angiographic results. The coronary artery progression is defined as atherosclerotic culprit lesion progression and clinical progression is defined as symptoms deterioration.

LIne 83: The progression was estimated by progression of previous mild stenosis, or a new stenosis and the requirement for percutaneous coronary intervention. It is not a way to evaluate angiographic progression. A progression of a mild stenosis could simply depend of a different operator who classified the stenosis in different way. It is well known that a single lesion could be classified differently by different operators or even by the same operator in different time

Dear Reviewer, thank you for suggestion but we can’t agree with you opinion as all the procedures were performed in one cath-lab center with the same team. The coronary artery lumen stenosis was estimated by % not as you implied by suggestive evaluation.

86.87 Participants with acute cardiovascular syndromes and those with congestive heart failure, hematological, oncological, thyroid, liver and kidney diseases or corticosteroid treatment were excluded from the analysis. It is strange to esclude patients with acute coronary disease. Isn't it a progression from a chronic condition?

Dear Reviewer, we want to thank you for your comment and the answer is enclosed in manuscript.

We present the relation between chronic coronary artery disease progression and annual exposure to air-pollution. Previous studies indicated the relation between acute coronary syndromes (ACS) and short-term exposure to air pollutants. In Jiang et al. [] analysis the greater admission risk from acute cardiovascular events was observed under high ozone pollution days. Dynamic process that occurs in acute coronary syndromes that involve plaque vulnerability, fibrinolytic function, and platelet activation responsible for acute event were found related to transient exposure to environmental air by Chen et al []. The relation between weather changes and increased risk for ST-segment elevated acute coronary syndromes were presented by Biondi-Zoccai et al. []. The impact of short-term air pollution in industrialized and non-industries areas on ACS was already found in Kuźma et al. analysis [].

118 electrocardiography (ECG) and transthoracic echocardiography (TTE) were performed in each patient before the procedures. Blood samples were collected before the procedures. Whole blood analysis was measured with routine hematology analyser (Sysmex Euro GmbH, Norderstedt, Germany). GFR was calculated by simplified Modification Diet in Renal Disease (MDRD) formula. Completely useless informations

Dear Reviewer, thank you for your valuable comments but we can’t agree with this suggestion. The ECG was essential as the blood samples to roule out the risk for acute syndromes. The whole blood count analysis was required to compare the obtained data with simple inflammatory indexes as NLR, MLR, SIRI et. as there is a magnitute of novel publication relating the coronary artery disease progression with aforementioned indexes. The GFR was essential to roule out the risk for coronary artery disease progression related to kidney dysfuntion.

Other risk factor? WHat about cholesterol? Were the findings normalised for cholesterol levels?

Dear Reviewer, the cholesterol results were taken into consideration and the aimed levels of LDL were reached.  The multivariable analysis reached all data regarding CV risk factors.

How was the rate of progression? If a lesion increase from 30 to 60% over a period of 5 years or from 80 to 90% over a period of 3 years what's worse?

Dear Reviewer,

I would like to thank you for your question, but the explanation is found in our manuscript. The progression was found significant while required PCI procedure. The initial artery lumen narrowing must be at least 50% to consider significant. On initial examination all culprit lesions >70% (>50% for LMCA) were stented, on repeated angiography, the significant progression (requiring repated PCI stenting) must be at least 70% (50% for LMCA). There was no chance to leave 80% coronary artery stenosis on initial exm,aination and wait for the progression to 90%. The 30% stenosis was found significant on repeated angiography if reached the level of minimum 70% stenosis.

Kind regards

Tomasz Urbanowicz

Round 3

Reviewer 1 Report

I've read with interest the third revision of the article. It has been improved a lot from the first version and english is much better and clear now.

I've appreciated the topic from the begin, but I still have my previous doubt which prevent me to suggest to accept the paper now, but I think it would be sorted easily. The title and the text speak about "coronary artery disease progression", but the authors are not able to use a method addressing coronary artery disease globally; they just look at single coronary lesions. If you don't look at global burden of coronary atheroslerosis you could not speak about coronary artery disease progression. I would suggest two possible solutions: the first one would be to look back to all the coronary angiograms of the patients and to calculate a SYNTAX score (modified excluding the points for collaterals and type of occlusion), which summarise the actual coronary artery disease (severity and diffusion). I know it would be a hard work and therefore I would suggest my second way out. To change in the title and in the whole text "coronary artery disease progression" with "coronary artery lesions progression"; in this way the authors' colcusions would be right and the paper could be accepted. I would prefer a global score for coronary artery disease extension and severity (therefore my first suggestion) but even the second would be ok.

Others small doubts:

3.1 has been all deleted? If so please rearrange numeration

Line 259: not clear. In Sielski et al. [45] analysis in acute coronary syndromes, the vascular access not the air pollutants were found the 260 key predictors of periprocedural deaths.

Some spelling errors throghout the text (example 262, 267 nitroc oxyde; 263 atherslerotic plaque, ...)

Minor spelling mistakes

Author Response

Poznan, September 11, 2023

Dear Reviewer,

We would like to express our high gratitude for taking your valuable time to review our manuscript as well as the highly encouraging comments.

In compliance with your professional suggestions, we have revised our manuscript as follows:  

  1. The title was modified.
  2. The corrections in manuscript were performed accordingly.

Please find all corrections in the manuscript marked in blue.

Once again, we would like to thank you for your help.

Kind regards

Tomasz Urbanowicz
